# Development and Validation of Empirical Models to Predict Metal Additively Manufactured Part Density and Surface Roughness from Powder Characteristics

**DOI:** 10.3390/ma15134707

**Published:** 2022-07-05

**Authors:** Paul Quinn, Sinéad M. Uí Mhurchadha, Jim Lawlor, Ramesh Raghavendra

**Affiliations:** 1SEAM Research Centre, Waterford Campus, South East Technological University, X91 TX03 Waterford, Ireland; sohalloran@wit.ie (S.M.U.M.); rraghavendra@wit.ie (R.R.); 2Department of Engineering Technology, Waterford Campus, South East Technological University, X91 K0EK Waterford, Ireland; jlawlor@wit.ie

**Keywords:** laser powder bed fusion, metal additive manufacturing, metal powder recycling, empirical modelling

## Abstract

Metal additive manufacturing (AM) processes, viz laser powder bed fusion (L-PBF), are becoming an increasingly popular manufacturing tool for a range of industries. The powder material used in L-PBF is costly, and it is rare for a single batch of powder to be used in a single L-PBF build. The un-melted powder material can be sieved and recycled for further builds, significantly increasing its utilisation. Previous studies conducted by the authors have tracked the effect of both powder recycling and powder rejuvenation processes on the powder characteristics and L-PBF part properties. This paper investigates the use of multiple linear regression to build empirical models to predict the part density and surface roughness of 316L stainless steel parts manufactured using recycled and rejuvenated powder based on the powder characteristics. The developed models built on the understanding of the effect of powder characteristics on the part properties. The developed models were found to be capable of predicting the part density and surface roughness to within ±0.02% and ±0.5 Ra, respectively. The models developed enable L-PBF operators to input powder characteristics and predict the expected part density and surface roughness.

## 1. Introduction

The ability to manufacture highly complex geometries and components is one of the many advantages of the metal additive manufacturing process Laser Powder Bed Fusion (L-PBF) [1]. In this process, shown in Figure 1, a metal powder is melted and fused selectively by a laser system in a layer-by-layer manner to build a component from the bottom up. The powder used in the L-PBF process is typically manufactured by atomisation and has a particle size in the range of 10 to 80 μm in diameter [2]. During the L-PBF process, the laser selectively melts regions of the powder bed; any powder that is not melted in this process can be re-used for subsequent builds using recycling and rejuvenation methodologies. This allows for greater utilisation of the costly powder material. Previous work by the authors has investigated the effect that these recycling and rejuvenation processes have on the powder characteristics and the resulting manufactured part properties [3]. It was found that the powder recycling process has a negative effect on the powder characteristics, in terms of degradation of the powder shape and mean particle size during the recycling process. These changes in the powder material were found to translate into changes in the resulting part properties, such as an increase in the surface roughness and increased porosity. These conclusions agreed with other studies across a range of different metal powders commonly used in the L-PBF process [4,5,6,7,8].

The resulting part quality from the L-PBF process is highly dependent on several factors, including the laser parameters, CAD geometries, and processing conditions, for example [9]. It is well understood that the characteristics of the raw input material, in this case a metal powder, affect the resulting qualities of the parts produced [1,3], research in previous years has reported the interdependent nature of the powder properties and resulting output parts manufactured [6,7]. Figure 2 conveys a simplified input-process-output diagram focused on the input raw material in the L-PBF process. The diagram displays the relationship between the powder characteristics and manufactured part properties. The ability to understand this relationship can enable greater utilisation of the metal powder within the L-PBF process.

Previous work by the authors [3] and other researchers [6,7,8,9,10] have provided a greater knowledge of the effects of powder recycling and rejuvenation processes, allowing L-PBF operators to improve utilisation of the powder material. Researchers have found that the overall powder recycling process leads to a degradation in the powder characteristics such as an increase in the particle size [10,11,12,13,14,15], a decrease in the circularity of the particles [9,10,11,12,15] as well as an increase in the oxygen concentration on the surface of the powder particles [9,14,16]. The combination of powder particle size and shape [17] influences the flowability of the powder material, which is an important characteristic in L-PBF due to the requirement for the powder to flow freely for the deposition of powder layers. As a result of the changes in the powder particle size and shape during powder recycling, the powder flowability improves [6,18,19]. The resulting manufactured part properties from the L-PBF process are dependent on the powder characteristics [3]. These changes in the powder characteristics have an effect on the resulting manufactured part properties such as increased surface roughness [3,6], increased porosity [17,18,19,20] and changes in the mechanical properties [6,17,18]. With this increased understanding of the effect of the powder recycling process, greater utilisation of the powder material used can be achieved.

Cordova et al. [8] presented an insight into how L-PBF part properties are affected by the powder characteristics throughout the recycling process. They suggested the use of a decision diagram to determine the outcome of the L-PBF process based on the input powder materials. This method provides a qualitative tool to indicate the expected outcome of the L-PBF process, using recycled powders.

The work presented in this paper sets out a series of empirical models that can be used to predict the manufactured part density and surface roughness of L-PBF parts based on the characteristics of the input powder materials that have been through a recycling and rejuvenation process. While previous researchers have provided an insight into the effect of powder recycling, this work provides an additional insight into the common powder rejuvenation process and its effect on the L-PBF process. This can assist in the development of a zero-waste and zero-defect manufacturing methodology for the L-PBF process.

## 2. Materials and Methods

### 2.1. 316L Stainless Steel Powder

The powder material used throughout this study is a 316L Stainless Steel, supplied by EOS GmBH (Krailling, Germany). This material is extensively used in the medical device and automotive industries due to its high corrosion resistance [4,5]. Table 1 outlines the chemical composition of the 316L powder used in this study.

### 2.2. Powder Characterisation

Prior to the development of the empirical models and relationships presented in this paper, the powder characteristics must be determined. This section will describe the methods used in this process.

To track the usage of the powder throughout the recycling and rejuvenation process, the accumulated build hours are used. This corresponds to the amount of time the powder is exposed to the L-PBF build process. For powder recycling, where no virgin powder is mixed into the batch, the build hours can be counted sequentially and used to represent the accumulated build time. For the rejuvenated powder, there is a combination of both virgin and recycled powder present within a batch. To calculate the accumulated build time on this batch of mixed powder, the Average Use Time (*AUT*), as proposed by Denti et al. [22] is used. The AUT is given by:(1)$$AUT\left(i\right)=\;\frac{q_{i}\left[AUT\left(i-1\right)+t_{i}\right]}{q_{o}}$$
where *q_i_* is the mass of virgin powder added, *AUT*(*i* − 1) is the number of build hours the recycled powder has been used for, *t_i_* is build time for *AUT*(*i*) and *q_o_* is the original mass of recycled powder material [22]. The AUT represents the Average Use Time of the powder batch considering the different use times for the virgin and recycled powder material. This allows for appropriate representation of the build hours for the powder material in this study.

The mean particle size (MPS) for each powder sample was determined by dry dispersion laser diffraction using a Microtrac S3500 Particle Size Analyser. This process disperses a sample of approximately 15 g of powder into a chamber in front of a tri-laser light source. Mie scattering theory is then used with resulting diffraction pattern from the particles crossing the light source is then used to extract the size of each particle in the sample [23] The recorded data are presented in the form of a percentage volume distribution against the particles size using a spherical model approximation. Samples were thoroughly mixed prior to loading into the test apparatus. The mean particle size was measured for three samples and averaged with a standard deviation calculated for each powder sample. This method allows for a large number of particles to be measured at once ensuring a statistically significant particle count is quantified [1].

To determine the effect of increased build cycles on the morphology of the powder particles, image analysis using ImageJ 1.53 k (New York, NY, USA) [24] was used on SEM images at ×200 magnification after each build. This imaging method allows for the powder morphology to be assessed both qualitatively and quantitatively in terms of observing the different powder shapes [1]. A powder sample was dispersed on a substrate for SEM imaging. The SEM images of particles were converted to binary images and then the area and perimeter of each particle were extracted, as shown in Figure 3. Between approximately 200 and 300 particles were analysed per powder sample.

The circularity, *C*, of each particle was calculated as follows:(2)$$C=4\pi \frac{A}{P^{2}}$$
where *A* is the area of the particle and *P* is the perimeter of the particles. A calculated circularity value of 1 represents a perfectly spherical particle [25]. Powder morphology heavily influences the flowability of the powder. Highly spherical powder material flows more freely than irregularly shaped powder material [16]. Previous research by the authors has shown that the increase in use of the powder material, i.e., powder recycling, results in the degradation of the circularity of the powder particles [3]. An average circularity, as calculated above, and standard deviation was determined for each sample of powder.

The skeletal density of the powder material is the density of powder in bulk form, representing the density of the sample of powder as a single consolidated mass. The skeletal density was determined by helium gas pycnometry using a Micrometics AccuPyc 1340 gas pycnometer. This is a displacement method in which a gas of a low density, i.e., Helium, is used to fill a chamber of known volume with a known mass of powder to a set pressure. The gas is then discharged into a second chamber of a known volume and the pressure of the gas in the chamber is measured, allowing for the volume of the powder sample to be calculated using Boyles Law [26]. The skeletal density provides an insight into the inter-particle porosity of the powder material; this can translate as porosity in the final component [27]. The pycnometry tests were repeated a five times per powder sample and then averaged and a standard deviation calculated.

### 2.3. Part Characterisations

Previous studies [3] conducted by the authors presented a series of test samples that were used to determine and quantify the effect of powder recycling on the as-built density and surface roughness of L-PBF parts. The samples, shown in Figure 4, were manufactured on an EOS M280 system with the default laser scanning parameters for 316L stainless steel powder, as shown in Table 2. The manufactured test samples, namely (a) Density Cube and (b) Overhang Block, enabled the authors to assess the impact of the powder recycling process on the output part properties.

The surface roughness of the overhang test specimen seen in Figure 4b was measured via focus variation microscopy, using a Bruker Alicona Infini Focus SL (Graz, Austria). Focus variation operates by capturing images of the surface at different focus heights and combining these images to compile a complete image of the surface utilising the areas of the images that are in focus [28]. The arithmetic mean, Ra, of the profile height deviation from a mean line over a given length was used to calculate the surface roughness Ra. Despite previous advances in aerial surface characterisation, the Ra parameter (ISO 4287) is the most widely adopted for characterising the roughness of metal AM surfaces [29]. Measurements were taken at ×5 magnification providing a vertical resolution of 460 nm and a measurement area of 13 mm^2^. Five areas of each surface were measured, and the Ra value was recorded and averaged with a standard deviation of the measurements recorded.

The part density of the density cube shown in Figure 4a was calculated based on image analysis of the micro-sections of these samples. Images of the polished micro-sections in different orientations enable the porosity, in terms of size, shape and quantity of the pores within the bulk material of the samples to be characterised and quantified [30]. Prior to imaging, the samples were prepared by mounting in epoxy resin in different orientations and then ground and subsequently polished to reveal micro-sections of the samples.

These micro-sections were imaged using a Keyence digital microscope VH-Z 100R (Milton Keynes, UK) at a magnification of ×200. Ten images of the micro-sections were taken to capture a representative cross-section of the sample. Images of the micro-sections were then imported into the ImageJ software [24], where the process outlined in Figure 5 was applied. This process consists of importing the microscope images into the ImageJ software, setting the scale of the image using the scale bar on the original microscope image. The image is then converted to a binary image, so the pores are presented as black spots on the micro-section. Finally, the pores are analysed. For each sample, 10 images were analysed using this method and the average porosity and standard deviation was determined for each sample. The percentage porosity present in the micro-sections is inversely proportional to the characterised density of the part. The material data sheet for 316L stainless steel states that as-built parts have a density of 100%, i.e., no porosity present [21].

### 2.4. Standard Error of the Mean

Each measurement of the characteristics was repeated multiple times to determine any uncertainty in the measured data. For all powder and manufacture part characterisations, the uncertainty in the measurement is represented by the standard deviation of the mean or the standard error of the mean (SE) as given by the following equation:(3)$$\mathrm{SE}=\frac{\sigma }{\sqrt{n}}$$
where *n* is the sample size (i.e., number of measurements taken) and *σ* is the standard deviation of the measurements [31]. For the developed models, the error is represented by the residuals generated. These residuals are corresponding the error between the measured value and model predicted value.

### 2.5. Multiple Linear Regression

Multiple Linear Regression (MLR) is a statistical method that uses several input variables to predict the outcome of a response variable. An MLR is used to determine the relationships between two or more variables [32]. An MLR is given by the following equation [33]:(4)$$Y_{i}=\beta_{o}+\beta_{1}x_{i1}+\beta_{2}x_{i2}+\cdots +\beta_{p}x_{ip}$$
where *i* is the number of observations, *Y_i_* is the dependent variable, *x_i_* is the explanatory variable or independent variable, *β_o_* is the *Y* intercept and *β_p_* is the slope coefficient for each of the explanatory variables. An MLR model is based on the following four assumptions [32], which must be tested once a model has been generated:There is a linear relationship between the dependent and independent variables.The independent variables must not be too highly correlated with one another.The observations are selected independently and randomly from the entire population.The residuals created from the model are normally distributed with a mean of zero and lie within one standard deviation of the mean.

The models were generated using the Multiple Linear Regression tool in Minitab^®^ Statistical Software, Version 20, (Coventry, UK). The empirical model was generated using 70% of the input data with the remaining 30% being withheld for validation of the empirical model. The 30% of data that was withheld was used to confirm the fit of the developed model with measured data.

Early studies of the relationships between powder characteristics and part properties indicated that part density and surface roughness were influenced greatly by the input powder characteristics. With this knowledge, an MLR model, for each of the part properties, part density and surface roughness, has been developed and validated in the following sections.

#### 2.5.1. Model Selection

Prior to the development of the empirical model, the range of variables to describe the dependent variables of part density and surface roughness were analysed in an exploration plot. This plot was completed to determine the strength of the relationship between the independent variables of the measured powder characteristics and the dependent variables. Each powder characteristic was plotted against the part density and surface roughness and the Pearson’s correlation coefficient extracted. This coefficient indicates the strength of relationship between the plotted values. A high Pearson Correlation coefficient, i.e., closer to 1, indicates a strong linear relationship between the powder characteristics and part properties explored. Those with a strong linear relationship shown in the scatter plots are likely to contribute significantly to MLR model; however, their significance to the developed model is tested further in the next stages of the model development.

#### 2.5.2. Model Testing

The first stage of testing the developed MLR models was to ensure that the inclusion of the selected independent variables contribute to the accuracy of the developed model. This is given by the test statistic, the *p*-value. If the *p*-value is smaller than 0.05, it that indicates the inclusion of the variable is statistically significant. Once all the variables to be included in the model were deemed significant, the coefficient of determination (R-Squared) was used to determine how much of a change in the dependent variable can be explained by the developed model. A higher R-Squared value indicates that more of the change in the dependent variable is explained by the model.

Testing the validity of the developed model is required to ensure that the selected model can represent the relationship with a high degree of accuracy and repeatability. To do this, the predicted values of the model were compared to the original measured values. The difference between these values is known as the residual. The testing of the model revolves heavily around these residuals and how they react with the independent variables used. These tests require four interactions to be analysed, as follows [32]:A plot of the residuals against the predicted values is normally distributed.The residuals are normally distributed.A plot of the residuals and the input variables (powder characteristics in this study) displays no significant trends.A plot of the residuals and the possible input characteristics omitted from the model displays no significant trend.

Once all the above conditions have been confirmed, the assumptions for the validity of an MLR model have been met, and the accuracy of the developed model can be established based on the largest residual generated.

## 3. Results and Discussion

### 3.1. MLR Selection

To determine the powder characteristics that influence the part density and surface roughness, a series of scatter plots of the independent variables (powder characteristics), against the dependent variables (part density and surface roughness) is required, as shown in Figure 6. These plots assist in the selection of the powder characteristics with the strongest linear relationship to the part density and surface roughness. The Pearson correlation coefficient, *r*, was also used to quantify the strength of this relationship between the plotted values. The variables with the highest Pearson correlations are then included in the development of the empirical model. These plots ensure that only those powder characteristics that are highly influential in the changes observed in the part property are included in the model development. These variables will be used in the development of the part density and surface roughness models.

As shown in Figure 6, the most influential characteristics for the Part Density are the AUT and Mean Particle Size, with Pearson coefficients of 0.756 and 0.700, respectfully. These characteristics agree with the work completed by Ahmed et al. [18], in which the authors observed an increase in the number of irregularly shaped particles as well as an increase in mean particle size, which exhibited a decrease in part density. In addition, Figure 6 highlights that the most influential characteristics for the part surface roughness are the AUT, Mean Particle Size and circularity with Pearson Coefficients of 0.975, 0.915 and 0.842, respectfully.

The empirical models presented in the following sections can be used to predict the part density and surface roughness resulting from the L-PBF process, for a 316L stainless steel powder on an EOS M280 machine. For the models to relate, the following constraints on the input powder characteristics are required:The AUT must be greater than 0 h.The powder circularity must be calculated as described in Table 1 and have a value of less than 1, indicating particles that are not perfectly circular in morphology.The mean particle size must be greater than 35 μm, as to ensure the laser parameters, shown in Table 2, are optimal for the powder used.

### 3.2. Part Density Model

As shown in Figure 6, the powder characteristics of AUT and MPS correlate highly with the part density and, as such, were included in the model development process. The MLR model generated with Minitab Statistical software for the prediction of part density is as follows:(5)Part Density= 100.466−0.000526AUT−0.01276MPS

Table 3 shows the summary statistics of the model developed. A low *p*-value for both the *AUT* and *MPS* indicates that their inclusion in the developed mode is significant. To further investigate the significance of the developed model, the R-Squared value is 98.79%; this value implies that the developed empirical model can explain 98.79% of any in the part density. The results presented in Table 2 confirm that the empirical model has been developed, and the significance of the variables included are confirmed.

The assumptions for MLR, as detailed in Section 2.5.2, are then tested. The results from these tests are shown in the graphs presented in Figure 7. As required for an MLR model, the residuals should be randomly distributed. Figure 7a shows a random scattering of the residuals centered around a mean of 0. Figure 7b confirms that residuals are normally distributed displaying a normal bell shape distribution curve for the residuals. Figure 7c–f are also plotted to ensure that the residuals from the model do not exhibit any significant trends with the variables selected for inclusion in the model and those omitted in the model selection phase. Figure 7c–f confirm that there is no significant trend, as shown by the random scattering of the plotted residuals between the selected or omitted model variables. These graphs once again confirm the summary statistics outlined in Table 3.

Figure 8 shows a comparison between the results from the developed model and the observed data. Figure 8 shows that the developed empirical model for the prediction of the part density based on the powder AUT and MPS can predict the final part density with an accuracy of ±0.02% of the part density.

The results presented in Figure 8 show that the model predicts that the density of the L-PBF samples decreased as the number of observations, in this case the number of recycling stages, increases. The model is capable of using the characteristics, AUT and Mean Particle Size, at each of these observations and predicting an expected part density to within ±0.02%. This confirms that the developed model can be used as a tool for determining the part density of L-PBF parts that are being manufactured with recycled powders.

### 3.3. Surface Roughness Model

Figure 6 conveys that the powder characteristics that correlated highly with the part density were the *AUT, MPS* and *Circularity*. The developed model including these variables is as follows:(6)Surface Roughness= −7.02 − 0.01274AUT − 12.31Circularity + 0.8537MPS

The summary statistics displayed in Table 4 are used to determine if the variables included in the model are statistically significant and to determine how much of a change in the surface roughness can be described by the model developed. As before, the low *p*-values for all the included variables confirms that their inclusion in the model is significant and therefore confirms their selection in the model, as previously shown in Figure 6. The presented R-squared value for the model indicates that 93.64% of the changes observed in the surface roughness measurement can be explained by the developed model, which accounts for the changing powder characteristics as a result of the powder recycling process.

Figure 9 shows the results of the MLR assumptions tests, as outlined in Section 2.5. The results show that the assumptions of an MLR model are met by the developed model. The requirement for the residuals to be normally distributed in order to satisfy the assumption for the model is shown in Figure 9a,b. This shows that the residuals are normally distributed with a mean cantered around 0. Further tests of the interactions of the model’s residuals and the included and omitted variables are shown in Figure 9c–f. These plots, once again due to a lack of a trend in these plots, it is confirmed that there are no interactions between the residues and the selected or omitted variables for the model. These tests confirm that the assumptions for the MLR model are achieved as well as verifying the models summary statistics shown in Table 4.

Figure 10 shows the relationship between the data emerging from the developed empirical model and the observed data. Figure 10 shows the developed empirical model developed to predict the surface roughness of the L-PBF samples based on the powder characteristics selected. As previously shown in Figure 6, the characteristic AUT, MPS and circularity displayed the strongest linear relationships with the changing surface roughness because of the powder recycling process. The model presented is capable of predicting the final part surface roughness with an accuracy of ±0.5 µm Ra.

The results presented in Figure 10 show that the developed model captures the increased surface roughness observed as a result of the powder recycling process. The model is capable of using the characteristics, AUT, Mean Particle Size and Circularity at each of these observations and predicting an expected surface roughness. These predicted values were found to be within ±0.5 Ra. Once again, as with the developed model for part density, the surface roughness model can be used as a tool for L-PBF operators to predict the outcome surface roughness from the characteristics of the recycled powder material.

## 4. Conclusions

This paper proposes two empirical multiple linear regression models for the prediction of the part density and surface roughness of manufactured components from the L-PBF process. The models predict the expected outcome for a 316L powder on an EOS M280 machine as the powder ages throughout the powder recycling process. The models utilise input powder characteristics including AUT, circularity and mean particle size, which can be determined prior to printing, to predict the manufactured part properties. A summary of the main findings are as follows:Scatter plots of the observed manufactured part properties against the powder characteristics of AUT, Mean Particle Size, Circularity and Skeletal Density were used to determine the most influential characteristics. The Pearson coefficient confirm that the powder Mean Particle Size and AUT produced the strongest relationship with the observed part density. The observed surface roughness had the strongest linear relationship with the AUT, mean particle size and circularity of the powder.The developed part density model met all the requirements for an MLR model and resulted in a model capable of predicting 98.79% of the changes observed in the part density.The part density model can predict the part density of the manufactured samples to within 0.02% of the observed values for the 316L powder that has been through the powder recycling process.The developed surface roughness model met all the requirements for an MLR model, resulting in a model that can predict 93.64% of the changes observed in the surface roughness.The surface roughness model can predict the manufactured part surface roughness to within 0.5 µm Ra of the observed values for the recycled 316L powder.

These developed empirical models will enable L-PBF operators to characterise the powder material prior to a build and use these quantified characteristics to estimate the expected manufactured part properties, namely part density and surface roughness. Further work will investigate developing similar models to enable the prediction of other output properties, such as mechanical and dimensional properties. This can enable a shift towards zero-defect L-PBF manufacturing.

## Figures and Tables

**Figure 1 materials-15-04707-f001:**
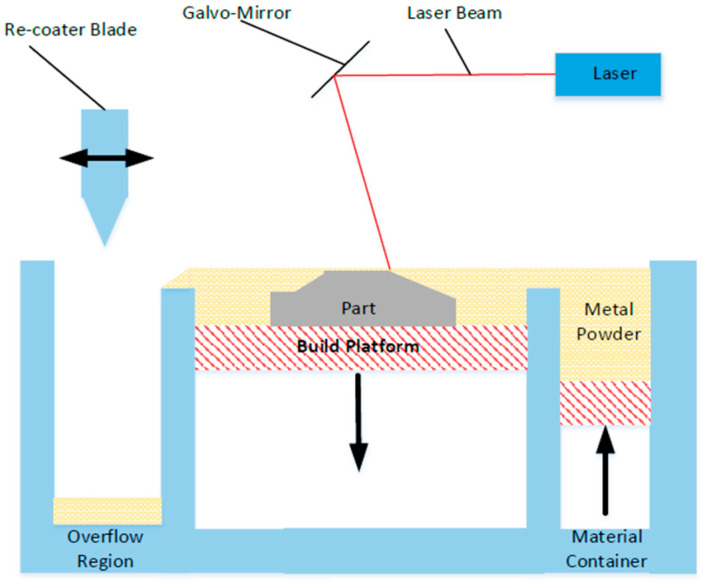
Laser Powder Bed Fusion (L-PBF) process.

**Figure 2 materials-15-04707-f002:**
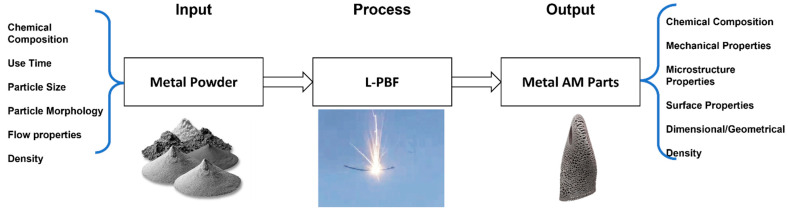
Input-Process-Output diagram for the LPBF process.

**Figure 3 materials-15-04707-f003:**
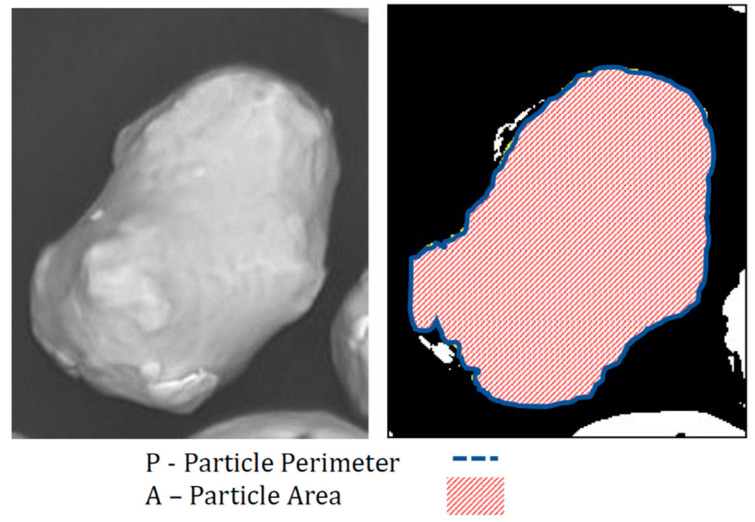
Powder Particle Circularity calculation description.

**Figure 4 materials-15-04707-f004:**
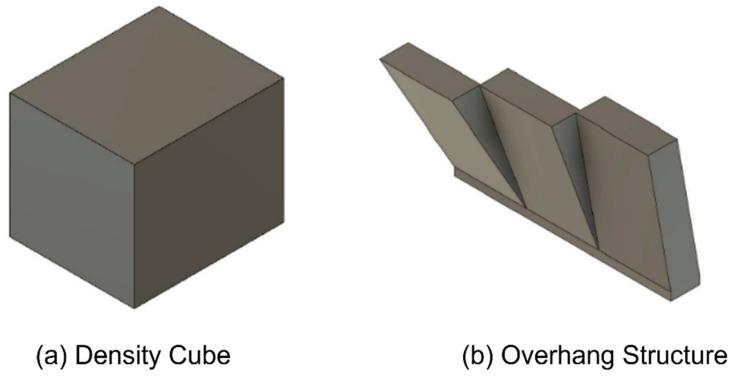
Manufactured test samples (**a**) Density cube for micro-sectional analysis of as-built parts and (**b**) Overhang structure to determine the surface roughness of.

**Figure 5 materials-15-04707-f005:**
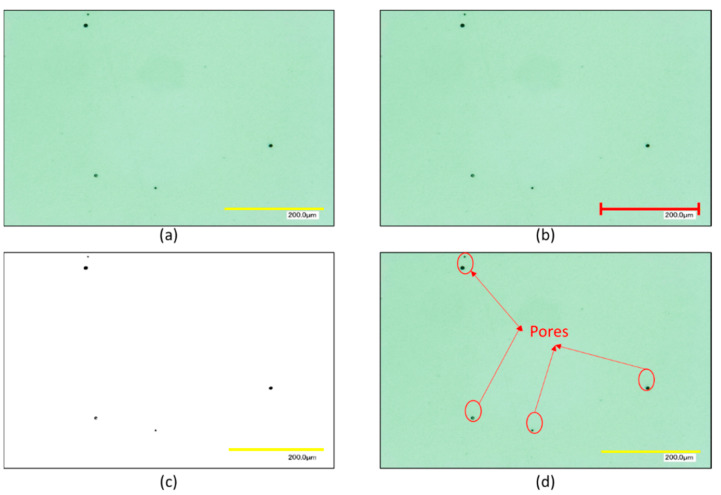
Part Density Measurement methodology (**a**) import microscope image, (**b**) set image scale, (**c**) convert image to binary and (**d**) analyse the pores present.

**Figure 6 materials-15-04707-f006:**
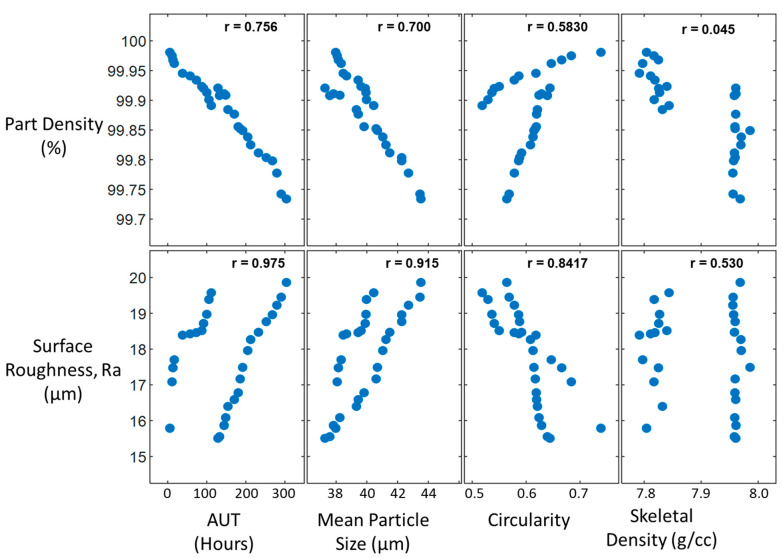
Scatter plots of independent variables against the dependent variables to determine those with the highest linear relationships.

**Figure 7 materials-15-04707-f007:**
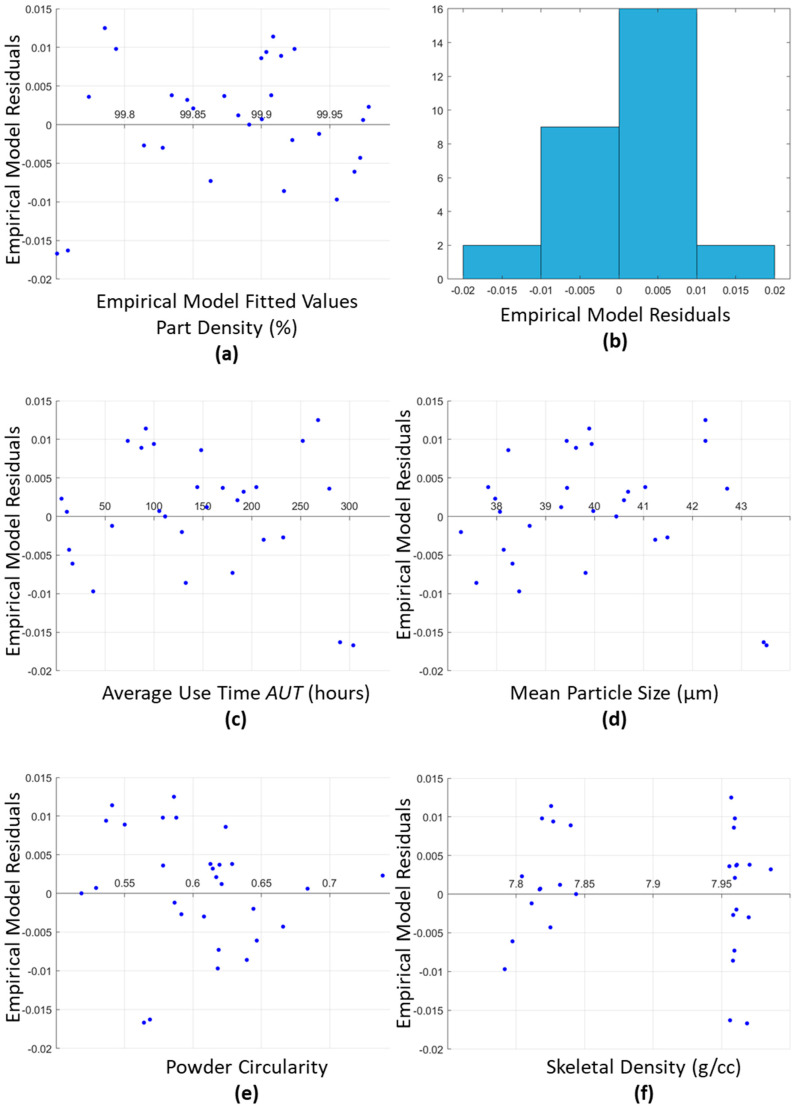
Part Density MLR Model results of the initial assumption tests to determine the validity of the model developed.

**Figure 8 materials-15-04707-f008:**
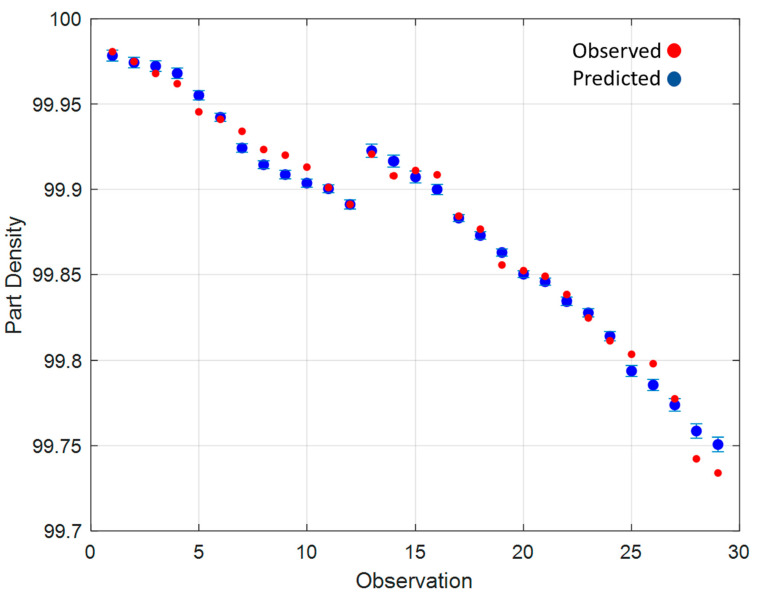
Part density model fitted values and observed part density values.

**Figure 9 materials-15-04707-f009:**
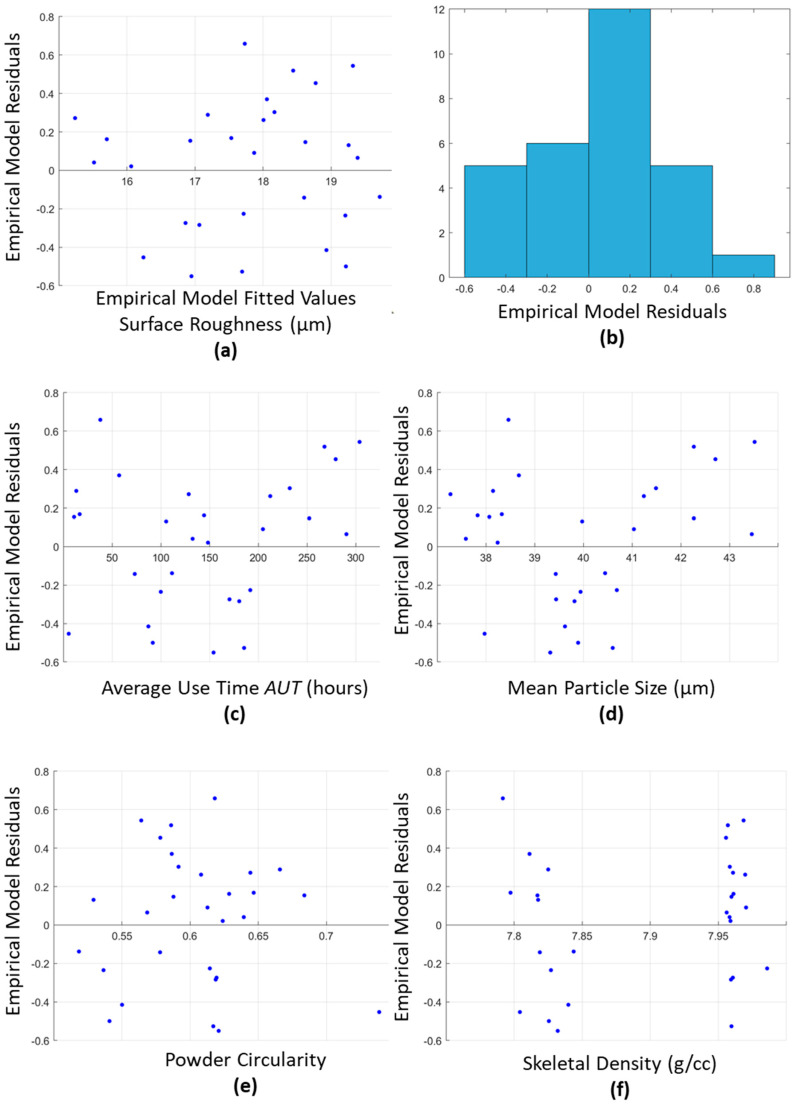
Surface Roughness MLR Model results based on the initial assumption tests, required to determine the validity of the developed model.

**Figure 10 materials-15-04707-f010:**
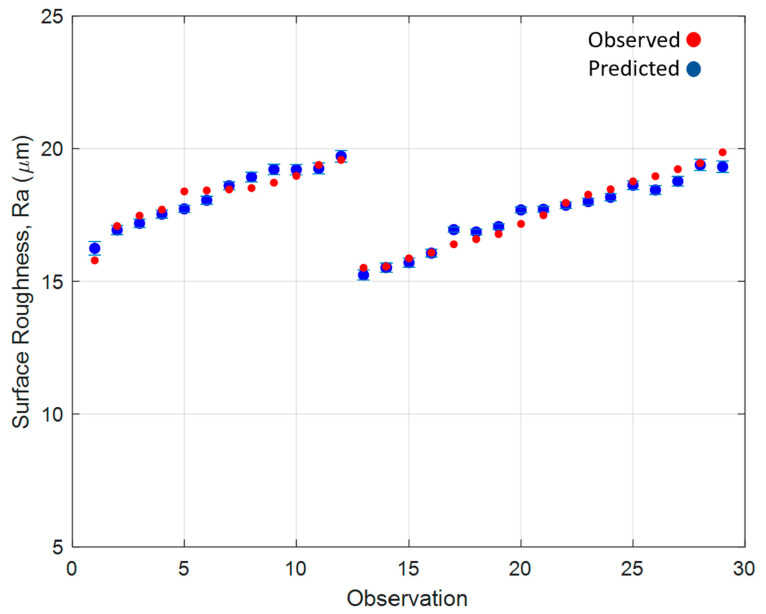
Surface roughness empirical model fitted values and observed values.

**Table 1 materials-15-04707-t001:** 316L Powder chemical composition as per Material Data Sheet [21].

Element	Mass (%wt)
Iron, Fe	Balance
Chromium, Cr	17.00–19.00
Nickel, Ni	13.00–15.00
Molybdenum, Mo	2.25–3.00
Carbon, C	0.03
Manganese, Mn	2.00
Copper, Cu	0.5
Phosphourus, Ph	0.025
Sulpher, S	0.01
Silicone, Si	0.75
Nitrogen, N	0.1

**Table 2 materials-15-04707-t002:** EOS Processing parameters for 316L Stainless Steel.

Parameter	Value
Laser Powder	195 W
Scan Speed	1000–1200 mm/s
Layer Thickness	20 µm
Hatch Spacing	90 µm
Build Platform Temperature	80 °C
O_2_ Concentration	0.1%

**Table 3 materials-15-04707-t003:** Summary statistics for the Part Density MLR model.

Source	R-Squared	*p*-Value
Part Density Model	98.79%	-
AUT		0.00
MPS		0.00

**Table 4 materials-15-04707-t004:** Summary statistics for the Surface Roughness MLR model.

Source	R-Squared	*p*-Value
Part Density Model	93.64%	-
AUT		0.00
MPS		0.00
Circularity		0.00

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
