# Peer review of "Development and Validation of Empirical Models to Predict Metal Additively Manufactured Part Density and Surface Roughness from Powder Characteristics"

_materials, 2022, doi:10.3390/ma15134707_

Round 1
Reviewer 1 Report
The paper submitted for review is titled: Development and validation of empirical models to predict metal additively manufactured part density and surface roughness from powder characteristics. This paper investigates the use of multiple linear regression to build empirical models to predict the part density and surface roughness of 316L stainless steel parts manufactured using recycled and rejuvenated powder based on the powder characteristics. However before the paper can be considered for publications, the authors are advised to amend the paper based on the comments below.
1. The overall language can be improved
2. Quality of images need to be enhanced. Avoid using plots with small fonts
3. Figure 4: It is not clear from the write up how these manufactured samples (different geometry) were considered in the regression model. What is the independent variable common to these geometries?
4. the authors are encouraged to discuss the limitation of this model and practical implications (how will this benefit the designer, etc.)
Author Response
The authors would like to thank the reviewers for dedicating their time and effort to read and review this work. This document summarises the changes that have been made to the paper in response to the reviewer comments. Each of the reviewers’ comments have been addressed separately. The changes made have are shown in red in the revised manuscript. The authors feel that the implementation of the reviewers’ comments have enhanced the quality of the submitted manuscript.
Reviewer 1 Author Responce
- The overall language can be improved
The manuscript has been reviewed and where necessary the overall language of the paper has been improved to ensure greater clarity for the reader throughout.
- Quality of images need to be enhanced. Avoid using plots with small fonts
The images and plots used in the manuscript have been updated to improve their clarity for the reader. High-quality images of each of the figures have also been included in the re-submission.
- Figure 4: It is not clear from the write up how these manufactured samples (different geometry) were considered in the regression model. What is the independent variable common to these geometries?
Figure 4 has been updated to highlight the samples that were relevant to the work presented in this manuscript. The following sections were added in order to clarify the aim and use of each of the samples shown in the figure. The authors feel that the changes made to the figure and the additional explanatory text clarify the contents of Figure 4.
“The manufactured test samples namely (a) Density Cube and (b) Overhang Block, enabled the authors to assess the impact of the powder recycling process on the output part properties.”
“The surface roughness of the overhang test specimen seen in Figure 4 (b) was measured via focus variation microscopy, using an Alicona Infini Focus SL.”
“The part density of the density cube shown in Figure 4(a) was calculated based on image analysis of the micro-sections of these samples.”
- The authors are encouraged to discuss the limitation of this model and practical implications (how will this benefit the designer, etc.)
A section regarding the benefits of the model to the L-PBF user has been added to the conclusions of the paper.
“These developed empirical models will enable L-PBF operators to characterise the powder material prior to a build and use these quantified characteristics to estimate the expected manufactured part properties, namely part density and surface roughness”
Reviewer 2 Report
Dear authors, a manuscript titled ‘Development and validation of empirical models to predict metal additively manufactured part density and surface roughness from powder characteristics’, Manuscript ID: materials-1636508, have some weakness that must be improved.
Please find some comments:
- Firstly, there is no paper referenced from the Materials journal. Reviewer totally agrees that there are no official requirements to search relevant papers in the journal where the manuscript is going to be published, however, even a few, e.g. two or three, sources should be indicated. This content is also relevant when selecting a suitable journal is processed.
- According to the first comment, furthermore, even the PBF technique is up-to-date, in reference, only 4 from 14 items are from the last 5 years. From that point of view, the manuscript issue seems to be out-of-date.
- Moreover, the ‘Introduction’ section, even suitable, shows no critical review of the current stage of knowledge. Further, when no lack of previous required studies is presented, the motivation of work could be lost. Please try to emphasize the novelty requirements.
- From my understanding, five different areas of the measured part were average to reduce the measurement uncertainty? Why five measurements of the same area were not measured and then averaged? Were there any limitations with relocation or other problems? It was not mentioned but significant.
- There is no word against both measurement uncertainty and measurement noise. Moreover, the selection of measuring equipment and, respectively, measurement technique, was not justified. Was there any measurement techniques suggested or, at least, addicted to the characterisation of the PBF details? Please look for some current (then a reference no. 12) papers considering comparison (and justification) of various measurement methods (techniques), measurement noise and uncertainty:
- https://doi.org/10.24425/123894
- https://doi.org/10.3390/ma14175096
- https://doi.org/10.1088/2051-672X/3/3/035004
- How were the profiles selected? Samples were measured with an areal (3D) performance but, respectively, parameter (Ra) was calculated for profiles (2D). It is not unified. Why a 3D, e.g. Sa or Sq, parameters were not calculated and, simultaneously, analysed?
- The section of results presented must be re-written that more figures and tables are commenting than a discussion around them. Please try to indicate some non-rough (raw) conclusions in this part.
- The ‘Conclusion’ section is too poor. Please, first of all, try to number each of the final suggestions separately, e.g. by numbered gaps. This may be crucial for presenting the novelty and final results more clearly. In its current form, it is difficult to be separate from the previous, already published, studies. Moreover, the conclusion must be presented in a more cosious way that makes the improvements of the manuscript results more visible.
Moreover, some editorial errors were found, as follows:
- There are many shortcuts and abbreviations in the manuscript that make the reader confused. Please create an additional section, e.g. ‘Abbreviation’, to make some simplifications in abbreviation specifications.
- In line 157 there, probably, should be located ‘:’ rather than ‘;’.
- In lines 201 and 202 at the end of a sentence, the dot should be found to be unified according to lines 204 and 206.
- In line 228, as in the previous (no. 9) case (comment) the ‘:’ probably would be preferred instead of ‘.’.
Generally, the submitted manuscript can be classified as interesting in that the presented results can be valuable in further research considering the PBF field of study. However, many issues make an understanding of the paper difficult and, simultaneously, the reader confused. Therefore, the manuscript must be re-worked and improve to reduce some weaknesses, before any further processing, if allowed.
Author Response
The authors would like to thank the reviewers for dedicating their time and effort to read and review this work. This document summarises the changes that have been made to the paper in response to the reviewer comments. Each of the reviewers’ comments have been addressed separately. The changes made have are shown in red in the revised manuscript. The authors feel that the implementation of the reviewers’ comments have enhanced the quality of the submitted manuscript.
1. Firstly, there is no paper referenced from the Materials journal. Reviewer totally agrees that there are no official requirements to search relevant papers in the journal where the manuscript is going to be published, however, even a few, e.g. two or three, sources should be indicated. This content is also relevant when selecting a suitable journal is processed.
The following articles from the Materials Journal which were found to be relevant to the content of the manuscript have been added to the reference list and reference in-text in the revised manuscript.
[2] J. Bedmar, A. Riquelme, P. Rodrigo, B. Torres, and J. Rams, “Comparison of different additive manufacturing methods for 316l stainless steel,” Materials (Basel)., vol. 14, no. 21, 2021, doi: 10.3390/ma14216504.
[6] R. Harkin et al., “Powder Reuse in Laser-Based Powder Bed Fusion of Ti6Al4V—Changes in Mechanical Properties during a Powder Top-Up Regime,” Materials (Basel)., vol. 15, no. 6, p. 2238, 2022, doi: 10.3390/ma15062238.
[7] R. Harkin, H. Wu, S. Nikam, J. Quinn, and S. McFadden, “Reuse of Grade 23 Ti6Al4V Powder during the Laser-Based Powder Bed Fusion Process,” Metals (Basel)., vol. 10, no. 12, p. 1700, 2020, doi: 10.3390/met10121700.
2.According to the first comment, furthermore, even the PBF technique is up-to-date, in reference, only 4 from 14 items are from the last 5 years. From that point of view, the manuscript issue seems to be out-of-date.
Several more current research articles that relate to the work presented in this manuscript have been added to the reference list and cited in-text. The overall quantity of the references included in the manuscript has increased. The total number of references from the previous 5 years is now 13 in the revised manuscript.
3. Moreover, the ‘Introduction’ section, even suitable, shows no critical review of the current stage of knowledge. Further, when no lack of previous required studies is presented, the motivation of work could be lost. Please try to emphasize the novelty requirements.
The introduction section has been updated to include a more critical review of the state-of-the-art in this research area. In this additional section, current methods of translating powder characteristics are presented and reviewed. The novelty of the method presented in this work is discussed.
“Researchers have found the overall powder recycling process leads to a degradation in the powder characteristics, such as an increase in the particle size [10–15], a decrease in the circularity of the particles [9–12, 15], as well as an increase in the oxygen concentration on the surface of the powder particles [9, 14, 16]. The combination of powder particle size and shape [17] influences the flowability of the powder material, which is an important characteristic in LPBF due to the requirement for the powder to flow freely for the deposition of powder layers. As a result of the changes in the powder particle size and shape during powder recycling, the powder flowability improves [6, 18, 19]. The resulting manufactured part properties from the L-PBF process are dependent on the powder characteristics [20]. These changes in the powder characteristics have an effect on the resulting manufactured part properties, such as increased surface roughness [6, 20], decreased porosity [19–21] and changes in the mechanical properties [6,19, 21]. With this increased understanding of the effect of the powder recycling process, greater utilisation of the powder material used can be achieved.
“Cordova et al. [8] presented an insight into how L-PBF part properties are affected by the powder characteristics throughout the recycling process. They suggested the use of a decision diagram to determine the outcome of the L-PBF process based on the input powder materials. This method provides a qualitative tool to indicate the expected outcome of the L-PBF process, using recycled powders.”
4.From my understanding, five different areas of the measured part were average to reduce the measurement uncertainty? Why five measurements of the same area were not measured and then averaged? Were there any limitations with relocation or other problems? It was not mentioned but significant.
All measurements were repeated for the characteristics presented in this work in order to ensure a greater representation of the powder characteristics and part properties of the L-PBF samples. Both multiple measurements of the same area and the part were completed in order to minimise this influence on the result. A mean value and standard deviation were extracted in order to assess the results.
All characterisation equipment was calibrated regularly throughout the study. This involved a series of measurements of a single area or samples. This included multiple measurements of a single area for both the surface roughness using various magnifications to ensure the recorded measurement was repeatable/ The part density measurements consisted of multiple measurements across a single cross section, this was then repeated for other cross sections to ensure that a representative area was characterised for the part density value.
5. There is no word against both measurement uncertainty and measurement noise. Moreover, the selection of measuring equipment and, respectively, measurement technique, was not justified. Was there any measurement techniques suggested or, at least, addicted to the characterisation of the PBF details? Please look for some current (then a reference no. 12) papers considering comparison (and justification) of various measurement methods (techniques), measurement noise and uncertainty:
- https://doi.org/10.24425/123894
- https://doi.org/10.3390/ma14175096
- https://doi.org/10.1088/2051-672X/3/3/035004
Measurement uncertainty for all of the collected results is represented by the Standard Error of the Mean. The following was added to address the inclusion of uncertainty in the measurements of the powder characteristics and part properties.
“Each measurement of the characteristics was repeated multiple times to determine uncertainty in the measured data. For all powder and manufactured part characterisations, the uncertainty in the measurement is represented by the standard deviation of the mean or the standard error of the mean (SE). The SE is given by the following equation:
SE=σ/√n (3)
where, n is the sample size (i.e. number of measurements taken) and σ is the standard deviation of the measurements [33]. For the developed models, the error is represented by the residuals generated. These residuals are corresponding to the error between the measured value and model-predicted value.”
6. How were the profiles selected? Samples were measured with an areal (3D) performance but, respectively, parameter (Ra) was calculated for profiles (2D). It is not unified. Why a 3D, e.g. Sa or Sq, parameters were not calculated and, simultaneously, analysed?
The Ra parameter was selected as this value had been previously extracted in the proceeding studies for the surface roughness of the parts. The authors agree that the Sa value provides a greater presentation of the overall surface roughness. The Ra value was selected as it remains the most widely adopted measurement for metal AM surfaces. The following text has been included in the manuscript to reflect this:
“Despite previous advances in aerial surface characterisation the Ra parameter (ISO 4287) is the most widely adopted for characterising the roughness of metal AM surfaces [31].”
7. The section of results presented must be re-written that more figures and tables are commenting than a discussion around them. Please try to indicate some non-rough (raw) conclusions in this part.
A series of sections have been added to in the results section of the manuscript which more critically assess the findings. These additional sections include:
“The results presented in Figure 8 show that the model predicts that the density of the L-PBF samples decreased as the number of observations increase, i.e. with increased recycling stages. The model is capable of using the powder characteristics, i.e. AUT and Mean Particle Size, at each of these observations and predicting an expected part density to within ±0.02%. This confirms that developed model can be used as a tool for determining the part density of L-PBF parts that are being manufactured with recycled powders. “
“The results presented in Figure 10 show that the developed model captures the increased surface roughness observed as a result of the powder recycling process. The model is capable of using the characteristics, AUT, Mean Particle Size and Circularity at each observation to predict the surface roughness. These predicted values were found to be within ± 0.5 µm Ra. Once again, as with the developed model for part density, the surface roughness model can be used as tool for L-PBF operators to predict the outcome surface roughness from the characteristics of the recycled powder material.”
8. The ‘Conclusion’ section is too poor. Please, first of all, try to number each of the final suggestions separately, e.g. by numbered gaps. This may be crucial for presenting the novelty and final results more clearly. In its current form, it is difficult to be separate from the previous, already published, studies. Moreover, the conclusion must be presented in a more cosious way that makes the improvements of the manuscript results more visible.
The conclusions section has been re-written to provide more clarity to the reader. A bullet list of the main findings is included with a section summarising the work completed as well as the main benefits of the work.
-
- Scatter plots of the observed manufactured part properties against the powder characteristics of AUT, Mean Particle Size, Circularity and Skeletal Density were used to determine the most influential characteristics. The Pearson coefficient confirmed that the powder mean particle size and AUT produced the strongest relationship with the observed part density. The observed surface roughness had the strongest linear relationship with the AUT, mean particle size and circularity of the powder.
- The developed part density met all the requirements for a MLR model and resulted in a model capable of predicting 98.79% of the changes observed in the part density.
- The part density model can predict the part density of the manufactured samples to within 0.02% of the observed values for the 316L powder that has been through the powder recycling process.
- The developed surface roughness model met all the requirements for an MLR model, resulting in a model that can predict 93.64% of the changes observed in the surface roughness.
- Surface roughness model can predict the manufactured part surface roughness to within an Ra value of 0.5 µm of the observed values for the recycled 316L powder.
9. There are many shortcuts and abbreviations in the manuscript that make the reader confused. Please create an additional section, e.g. ‘Abbreviation’, to make some simplifications in abbreviation specifications.
The use of abbreviations in the manuscript corresponds with the Journals instructions to authors which states that abbreviations should be defined only at their first time of use in the text. To keep in-line with the author guidelines for the Materials Journal an Abbreviations section has not been included in this revision.
10.In line 157 there, probably, should be located ‘:’ rather than ‘;’.
11.In lines 201 and 202 at the end of a sentence, the dot should be found to be unified according to lines 204 and 206.
12.In line 228, as in the previous (no. 9) case (comment) the ‘:’ probably would be preferred instead of ‘.’.
Comments 10 through to 12 have been addressed in text by the authors.
Round 2
Reviewer 2 Report
Dear authors, manuscript ‘Development and validation of empirical models to predict metal additively manufactured part density and surface roughness from powder characteristics’, Manuscript ID: materials-1636508, has been improved in a required manner so, respectively, can be further processed.
Thank you for your responses that, in their current form, were addressed properly and make the manuscript more suitable for publication in a quality journal as the Materials is.